



Notes on the correlation between SSWs and solar activity
Ekaterina Vorobeva (st062133@student.spbu.ru)
Department of Atmospheric Physics, Saint-Petersburg State University, Universitetskaya
Emb. 7/9, 199034, Saint-Petersburg, Russia
**Abstract**
A correlation between solar activity and normalized occurrence rate of sudden stratospheric
warmings (SSWs) has been found. As a proxy for solar activity, F10.7 cm radio flux has been
used. In order to find the correlation, we derived a normalized occurrence rate of MSSWs
based on both ERA40/ERA-Interim dataset and NCEP data. Based on this distribution, we
calculated the correlation coefficient, which amounts to 0.6314, with a significance of 90.68%
for ERA40/ERA-Interim, and 0.5455 for NCEP-NCAR-I, with a significance of 83.80%.
Additionally, we calculate correlation coefficients for Lyman-alpha flux and sunspot numbers
with the analogous method for the same period.
Keywords: Middle atmosphere – composition and chemistry; Waves and tides; Middle
atmosphere dynamics

**1. Introduction**

In the middle of the last century, Scherhag (1952) and Scrase (1953) independently found an
incident of sudden stratospheric warming (SSW). A corresponding mesospheric cooling has
been found shortly after (Quiroz, 1969). The SSW effect is manifested in sudden and short
(several days) increase in temperature (up to 60 K) in stratosphere and joint cooling in the



mesosphere at high and middle latitudes during winter. More strict definition of SSW one can
find in any review on this subject (e.g. Butler et al., 2015). According with current knowledge
(see e.g. Shepherd et al., 2014; Zülicke et al., 2018; and references therein) the genesis of the
effect goes from mesopause at high latitudes toward stratosphere at middle latitudes with peak
of intensity around 65° N. There are two types of sudden stratospheric warmings: minor
warmings and major warmings. Minor warmings also consist of the temperature increase, but
at 10 hPa it is about 30 K smaller than for major warmings. The main difference is that unlike
to the major warming, during the minor one, the zonal wind weakens but does not reverse the
direction (e.g.  Labitzke, 1981). In this study, we consider just major sudden stratospheric
warming effect.
SSW events play a rather important role in atmospheric investigations not only because these
pronounced events have impacts on all processes in the middle atmosphere but also because
they provide a natural examination of our understanding of atmospheric interactions. The first
step to understanding the nature of SSWs was the theory of planetary waves (PWs)
propagation by Charney and Drazin (1961), who derived the dispersion relationship for
vertically propagating Rossby waves. The theoretical explanation was proposed by Dickinson
(1968a,b; 1969a,b) and consists of an interaction of PWs which penetrate into the winter
middle atmosphere and affect general mean circulation when they dissipate. Steady
dissipating waves can weaken the zonal mean flow and maintain the winter stratosphere
above radiative equilibrium temperatures (Dickinson, 1969b). This theory was confirmed by
model simulations (Matsuno, 1970, 1971). Currently, this explanation is generally accepted;
nevertheless, we should note that there are alternatives. For example, based on model
simulations, Peters (1985 a,b) found that SSW-like effects may occur due to nonlinear wave–
wave interactions. However, the role of wave–wave interaction during SSWs is not clear until
the present time. Recently, Gavrilov et al. (2017) have touched upon this problem.

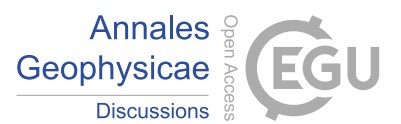

Since SSWs have been observed and modeled in numerous works (e.g. Holton, 1976;
Schoeberl, 1978; Tao, 1994; Siskind et al., 2005; Smith et al., 2011, and references therein),
the topic has attracted genuine interest in all fields of atmospheric science. Using a 3D model,
Sonnemann et al. (2006) studied the distributions of minor chemical species in the mesopause
region in time of SSWs. The most-detailed investigation of the variability of the hydroxyl
airglow layer during SSWs has been represented in the work of Shepherd et al. (2010). The
response of OH* and the infrared atmospheric band has been found by satellite observations
(Gao et al., 2011), and Shepherd et al. (2014) investigated the impact of this phenomenon on
distributions of CO and $NO_x$ based on a joint analysis of model simulation and satellite
observations. The impact of SSWs on the secondary ozone layer has been highlighted in the
work of Tweedy et al. (2013) based on model simulations and in Smith et al. (2009) based on
the SABER instrument onboard the TIMED satellite. The temperature and dynamic structure
of the mesopause region during sudden stratospheric warmings were investigated by
reanalysis data (Siskind et al., 2010) and based on a global circulation model by Zülicke and
Becker (2013). A large number of works are devoted to the role and propagations of gravity
waves in times of SSWs (Limpasuvan et al., 2011, 2012; McLandress et al., 2012; de Wit et
al., 2014; Ern et al., 2016). Recently, an effect on the troposphere (Hinssen et al., 2011) and
equatorial latitudes has been found (Bal et al., 2017). More about SSWs and related fields can
be found in reviews of this subject (e.g. Holton, 1980; McIntyre, 1982; Plumb, 2010; Butler et
al., 2015).
One of the strongest effects on the nature of Earth comes from the sun (Seppälä et al., 2014);
hence, naturally, the question of what the effect of solar variations on the SSW occurrence
rate arises. The strongest solar variation is the 11-year solar cycle. Labitzke and van Loon
(1990) did not find any significant correlation between the 11-year solar cycle and MSSWs
based on their analysis of F10.7 flux. Nevertheless, Labitzke (2004, and references therein)
showed that such a correlation exists for MSSW events distributed by phases of QBO (quasi



biennial oscillation). This is partially in contradiction with work of Sonnemann and
Grygalashvyly (2007), who found such a correlation without a relationship to QBO phases
based on an analysis of Lyman-alpha flux and sunspot numbers. The reason for the
discrepancy is either the difference in fluxes or methods.
We decided to narrow this gap in the knowledge and conduct an analysis of the solar radio
flux at 10.7 cm (F10.7 flux). However, based on SSW statistics and F10.7 radio flux, we
derived a normalized occurrence rate for MSSW events. The data, method, and results are
described in Sect. 2, followed by concluding remarks in the last section.

**2. Data, Method, and Result**

We investigate the statistical connection between MSSWs and solar activity. As a proxy for
solar activity, we use F10.7 radio flux (http://lasp.colorado.edu/lisird/data/noaa_radio_flux/).
Because MSSWs are phenomena that commonly occur from December until March (Charlton
and Polvani, 2007), we calculated monthly mean values of F10.7 radio flux for December,
January, February, and March through the entire period from 1958 to 2013. The lowest mean
F10.7 radio flux value did not fall below 67 solar flux units (sfu). The uppermost value did
not exceed 267 sfu. We chose a difference of 25 sfu for the flux subdivision (8 subintervals)
and calculated a number of monthly mean F10.7 radio flux values which fell into each
subinterval (Fig. 1a).
Next, we calculated the mean F10.7 flux values for the month prior to the MSSWs' central
day (the day when zonal mean zonal wind at 10 hPa becomes negative). In this study, we used
two databases of central day. The first database combines the central day of MSSW events
from ERA-40 reanalysis for the period 1958 to 1979 and ERA-interim reanalysis for the
period 1979 to 2013 (Butler et al., 2017). The central days by NCEP-NCAR-I reanalysis
(Butler et al., 2017) were used as the second database. Then, we calculated the number of

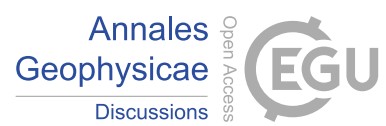



MSSWs that occurred in each F10.7 radio flux subinterval (Fig. 1b) based on two databases of
central day. The dependence of MSSWs on F10.7 flux is rather negative (Fig. 1b), but we
should take into account that the distribution of wintertime monthly averaged values of F10.7
flux is non-uniform. The values corresponding to low solar activity occur most often, and
values corresponding to high solar activity are rare. Hence, for calculations of correlation
between MSSW and F10.7, MSSW occurrence rate should be normalized. We calculated the
MSSWs' occurrence rate normalized to the occurrence rate of F10.7 flux values as shown in
Sonnemann and Grygalashvyly (2007):
$$R^i = \frac{\left(N^i_{MSSW} \middle/ N^i_{F10.7}\right) \sum N^i_{MSSW}}{\sum \left(N^i_{MSSW} \middle/ N^i_{F10.7}\right)}, \quad i = 1,...,8, \tag{1}$$

where $N^i_{F10.7}$ and $N^i_{MSSW}$ are the number of F10.7 flux values and number of MSSWs in
subinterval $i$, respectively.
Fig. 1c illustrates dependence between the normalized occurrence rate of MSSWs and the
values of F10.7 flux according to Eq. (1) for ERA and NCEP-NCAR-I databases. We
conducted the correlation analysis for the normalized occurrence rate of MSSWs and the
F10.7 flux values with 8 subdivisions (Fig. 1d). The correlation coefficient equals 0.6314 for
the ERA case and 0.5455 for the NCEP-NCAR-I case. The significance amounts to 90.68%
and 83.80% for ERA and NCEP-NCAR-I, respectively. The results demonstrate a distinct
statistical connection between the normalized MSSW events and the F10.7 flux values. Our
correlation coefficients are smaller than those of Sonnemann and Grygalashvyly (2007),
probably, because we use different periods.
It is not the aim of this contribution to discuss consequences and reasons, but a possible
explanation for the correlation is the impact of solar activity either on PWs strength and
activity or on propagation conditions (e.g. Arnold and Robinson, 1998; Fröhlich and Jacobi,
2004). Recently, Koval et al. (2018) found that solar activity might affect meridional




temperature gradients and consequently change the vertical structure of the zonal wind and
PWs' propagation conditions. This may point to a potential explanation. Another one
possibility to explain obtained correlation is the interaction of cosmic rays (which anti-
correlate with solar activity) with atmosphere, and, particularly, with stratosphere, and have
an impact on climate (see Fig. 7 in Usoskin (2017) and corresponding discussion).
The F10.7 radio flux differs by the nature from the Lyman-alpha flux and sunspot numbers
(Bruevich et al., 2014; Mei et al., 2018). Thus, the information about correlation coefficients
for the same database and method potentially can be useful to identify possible reasons of
correlation. Hence, such correlation coefficients with corresponding significance are
calculated and stored in the Table 1.

**3. Summary**

We investigated the statistical relationship between solar activity and occurrence rate of major
sudden stratospheric warmings (MSSWs). For this purpose, F10.7 radio flux has been used as
a proxy for solar activity. The calculations have been performed based on two datasets of
central day (NCEP-NCAR-I and combined ERA) for the period from 1958 to 2013. The
analysis of calculations was based on the normalized MSSW occurrence rate. The analysis
revealed a positive correlation between MSSW events and solar activity with a correlation
coefficient equals 0.6314 for the ERA case and 0.5455 for the NCEP-NCAR-I case. Note that
the correlation is necessary but not a sufficient condition for a relationship between the two
phenomena. The nature of the correlation is still not clear, and further investigations in this
direction are necessary.







**Data availability.**
The F10.7 and Lyman-α solar flux data are available at http://lasp.colorado.edu/lisird/. The
sunspot numbers data are accessible at https://www.ngdc.noaa.gov/stp/solar/ssndata.html.

**Acknowledgements.**

The author is grateful to her teachers Prof. Dr. V. A. Yankovsky, Prof. Dr. G. Sved, and Prof.
Dr. E. L. Genikhovich.

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



**Tables.**

Table 1. Values of the correlation coefficient between solar activity and MSSWs for different

proxies. The number of subintervals is the same for all calculations.

| | F10.7 radio flux | | American Sunspot numbers | | Lyman-alpha flux | |
|---|---|---|---|---|---|---|
| ERA40/ERA-Interim | 0.6314 | | 0.5780 | | 0.5408 | |
| | | 90.68% | | 86.66% | | 83.36% |
| NCEP-NCAR-I | 0.5455 | | 0.4879 | | 0.5770 | |
| | | 83.80% | | 78.00% | | 86.57% |

**Figures.**

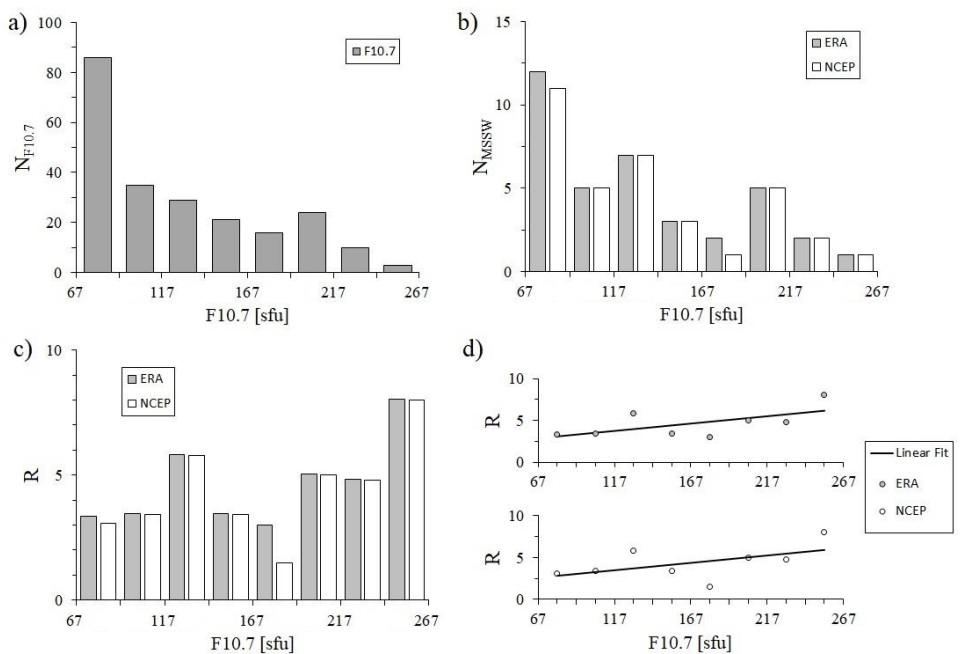

**Figure 1.** a) Monthly mean F10.7 flux values between 1958 and 2013 of 4 months between

December and March; b) the number of MSWWs depending on F10.7 flux values; c)

normalized occurrence rate of MSSWs depending on F10.7 flux values; d) correlation

analysis for normalized occurrence rate of MSSWs and F10.7 flux values.