# Peer review of "Notes on the correlation between SSWs and solar activity"

_Annales Geophysicae, 2019_

## Referee Comment (RC1) · Anonymous Referee #1 · 8 Apr 2019

The short paper deals with an old problem: is there an influence of the solar activity cycle on meteorological processes in the middle atmosphere such as major sudden stratospheric warming (MSSW)? In contrast to Sonnemann and Grygalashvyly 2007 (S&G), Labitzke did not find any clear correlation between the occurrence rate of MSSW and solar activity. Sonnemann and Grygalashvyly used the Lyman-alpha radiation and the sunspot number as proxy for solar activity. The Lyman-alpha radiation dissociates such constituents as molecular oxygen, water vapour or ozone, etc. The wavelength of the Lyman-alpha line lies in a deep absorption window of $O_2$. Its radiation penetrates down to the mesosphere. The optical depth of unity for overhead sun ranges approximately at 75 km, meaning there is an influence of the solar activity on the mesospheric chemistry.

[Figure]

However, a direct correlation between the occurrence rate of MSSW and solar activity within the regarded time interval of some (5) solar activity cycles is misleading, because it does not consider that the atmospheric system stays differently long in the arbitrarily defined (25 solar flux units sfu) solar activity bins. So generally, the system remains longer under low activity condition than under high activity conditions (see Figure 1b). The clue of the paper consists just in it to regard the variable long duration of the solar activity in the respective bins. In the paper this fact was called "normalized occurrence rate". The whole activity range between 67 and 267 sfu was divided into an integer number (8) of bins which contains the number of realization of solar activity belonging to the respective bin. As a proxy for solar activity the solar 10.7-cm-flux (called F10.7) was used. Additionally, the sunspot number and the Lyman-alpha radiation were employed as proxy for solar activity. There are, of course, further proxies for solar activity. Compared with the paper of S&G the number of bins is different, other solar parameters were used, and the time series was prolonged. The findings of S&G were on principle confirmed. The correlation was best for Lyman-alpha as proxy for solar activity, but the derived correlation coefficient was somewhat smaller than found in S&G.

I recommend this paper to publish in Annales Geophysicae after some changes listed below.

A positive correlation between MSSW and F10.7 is a statistical result which does nothing state about the mechanism of connection. In the paper few speculation about possible causes are presented. There occurs a possible bias due to decreasing strength of the solar cycles (from cycle 21 to cycle 24 now) and the simultaneous increasing cooling of the middle atmosphere due to growing $CO_2$ concentration (e.g. Berger and Lübken, 2011) and a general trend in stratospheric ozone by increase of the concentration of some minor constituents such as methane, N2O and other greenhouse gases. This entails a trend in the composition independent of solar activity. Also a variation in the ozone concentration over a solar cycle (Keating et al., 1987; Hartogh et al.,

2011) could influence the occurrence rate of MSSW by changing of the thermal structure of the middle atmosphere. For instance, Labitzke (2001) found an anti-correlation between F10.7 and temperature the middle and high latitudes.

Please define and explain in more detail the expression "normalized" (line 109).

Chapter 2 should be split inserting Chapter 3 "Discussion" after line 123. Summary is then Chapter 4.

Figure 1a seems to indicate that the occurrence rate of MSSW is inverse proportional to the F10.7 flux. Figure 1b explains why this is not right. Using Equation 1 one gets Figure 1c. This figure is the basis to calculate Figure 1d and to derive some statistical assertions. However, it should be mentioned that already the step from Figure 1b to 1c entails a statistical uncertainty which decreases with the number of solar cycles.

The references Kouker and Brassseur, 1986; Labitzke, 1987 and 2001; Labitzke et al., 2006; Liu et al., 2002; Tapping, 2013 and van Loon and Labitzke, 2000 are missing in the Text. (It is not necessary to quote Labitzke so often, your paper deals with the influence of the F10.7 flux upon the occurrence rate of MSSW, not with the connection between the occurrence rate of MSSW and the phase of the QBO.)

Authors beginning with Sh. . . should be quoted after Sc. . . in the list of references (e.g. Shepherd after Scherhag ).

The reference Charlton et al., 2007 is double. Line 91: Charlton et al., 2007.

Line 24/25: A corresponding mesospheric cooling has been found shortly after. The SSW starts with a mesospheric cooling before the SSW occurs in the stratosphere.

Line 72 What is meant with: "One of the strongest effects on the nature of Earth comes from the sun. . ."?

Line 78/80. . .without to consider a relation to QBO. . .

Line 123 Not only: "different periods", but also different bins, different solar proxies.

Keating et al. 1987, reference see in Sonnemann and Grygalashvyly, 2007.

Hartogh, P., G.R. Sonnemann, M. Grygalashvyly, and Ch. Jarchow, Ozone trends in the mid-latitude stratopause region based on microwave measurements at Lindau (51.66° N, 10.13° E), the ozone reference model, and model calculations, 2011, Adv. Space Res. 47, 1937-1948. https://doi.org/10.1016/j.asr.2011.01.010.

Berger, U., and F.-J. Lübken (2011), Mesospheric temperature trends at mid-latitudes in summer, Geophys. Res. Lett.,38, L22804,doi:10.1029/2011GL049528.

Please also note the supplement to this comment:
https://www.ann-geophys-discuss.net/angeo-2019-21/angeo-2019-21-RC1-supplement.pdf
* * *

---

## Author Comment (AC1) · 27 Apr 2019

Dear Referee, Thank you a lot for your constructive suggestions. We tried to follow your comments and suggestions.

Specific comments.

Referee writes: "There are, of course, further proxies for solar activity." In order to satisfy the referee and to enlarge an area of paper's application, we add four proxies (solar 3.2 cm, 8 cm, 15 cm, and 30 cm fluxes) in Table 1.

Referee note: "A positive correlation between MSSW and F10.7 is a statistical result which does nothing state about the mechanism of connection." We have similar notation in the Summary section, i. e.: "Note that the correlation is necessary but not a

sufficient condition for a relationship between the two phenomena".

Referee notes: " There occurs a possible bias due to decreasing strength of the solar cycles (from cycle 21 to cycle 24 now) and the simultaneous increasing cooling of the middle atmosphere due to growing CO2 concentration (e.g. Berger and Lübken, 2011) and a general trend in stratospheric ozone by increase of the concentration of some minor constituents such as methane, N2O and other greenhouse gases. This entails a trend in the composition independent of solar activity" The separation of the effects of long-term changes in solar cycle and long-term changes of anthropogenic greenhouse gases (GHGs) and ozone-depleting substances (ODSs) on the middle atmosphere still remains unsolved problem. Yes, generally speaking, joint declining of solar cycle and growth of GHGs and ODSs may produce bias in correlation. But according with current knowledge, there is no statistically significant impact of anthropogenic changes on frequency of SSWs (e. g. Butchart et al., 2000; SPARC CCMVal, 2010; Mitchell et al., 2012; Hansen et al., 2014, Ayarzagüena et al., 2018). Moreover, some of recent works show increase of the SSWs frequency (e.g., Huebener et al., 2007; Charlton-Perez et al., 2008; Bell et al., 2009; Schimanke et al., 2013; Ayarzagüena et al., 2013). Thus, in last case, the join effect of negative trend in solar cycle strength and positive trend of GHGs may just reduce positive correlation, but cannot be its cause. We add similar notation into the section Discussion.

Referee writes: "Please define and explain in more detail the expression "normalized" (line 109)." We rewrote line 109 in order to explain the expression "normalized" used in the text. Due to the limitation of paper size, we do not describe in detail a process of using a norm factor but we present the reference where one can find it.

Referee writes: "Chapter 2 should be split inserting Chapter 3 "Discussion" after line 123. Summary is then Chapter 4." Chapter 2 was split into Chapter 2 "Data, Method, and Result" and Chapter 3 "Discussion". In addition, we expanded Chapter 3 "Discussion" according to the referee's comments and suggestions.
Referee writes: "However, it should be mentioned that already the step from Figure 1b to 1c entails a statistical uncertainty which decreases with the number of solar cycles." We noted this fact right after the equation (1).

Referee writes: "The references . . . are missing in the Text. (It is not necessary to quote Labitzke so often, your paper deals with the influence of the F10.7 flux upon the occurrence rate of MSSW, not with the connection between the occurrence rate of MSSW and the phase of the QBO.)" Thank you for this remark. We removed the references missing in the text.

Referee writes: "Authors beginning with Sh. . . should be quoted after Sc. . . in the list of references (e.g.Shepherd after Scherhag )." Thank you for this remark. We rewrote the list of references in alphabetical order.

Referee writes: "The reference Charlton et al., 2007 is double. Line 91: Charlton et al., 2007." The reference in Line 91 was changed to Charlton et al., 2007.

Referee writes: "Line 24/25: A corresponding mesospheric cooling has been found shortly after. The SSW starts with a mesospheric cooling before the SSW occurs in the stratosphere." Currently, there are no unique opinion on time delay between SSW and mesopause cooling. Some authors state that they coincide (e. g. Zülicke et al., 2018). We do not touch this question in our short note and do not want make any strong statements on this subject.

Referee writes: "Line 72 What is meant with: "One of the strongest effects on the nature of Earth comes from the sun. . ."?" The author wanted to notice the solar influence on the Earth's atmosphere. Line 72 was rewritten to clarify the point.

Referee writes: "Line 78/80. . .without to consider a relation to QBO. . ." Corrected according to the reviewer's comment.

Referee writes: "Line 123 Not only: "different periods", but also different bins, different solar proxies." We added other possible reasons for the difference of correlation

coefficients.

Thank you a lot for taking the time to review the manuscript.

With respect, Ekaterina Vorobeva.

Please also note the supplement to this comment:
https://www.ann-geophys-discuss.net/angeo-2019-21/angeo-2019-21-AC1-supplement.pdf